# Behavioral and Psychiatric Disorders in Syndromic Autism

**DOI:** 10.3390/brainsci14040343

**Published:** 2024-03-30

**Authors:** Ann C. Genovese, Merlin G. Butler

**Affiliations:** Department of Psychiatry and Behavioral Sciences, University of Kansas Medical Center, Kansas City, KS 66160, USA; mbutler4@kumc.edu

**Keywords:** autism spectrum disorder, syndromic autism, gene, chromosome, syndromes, neurodevelopment, behavior, psychology, psychiatry

## Abstract

Syndromic autism refers to autism spectrum disorder diagnosed in the context of a known genetic syndrome. The specific manifestations of any one of these syndromic autisms are related to a clinically defined genetic syndrome that can be traced to certain genes and variants, genetic deletions, or duplications at the chromosome level. The genetic mutations or defects in single genes associated with these genetic disorders result in a significant elevation of risk for developing autism relative to the general population and are related to recurrence with inheritance patterns. Additionally, these syndromes are associated with typical behavioral characteristics or phenotypes as well as an increased risk for specific behavioral or psychiatric disorders and clinical findings. Knowledge of these associations helps guide clinicians in identifying potentially treatable conditions that can help to improve the lives of affected patients and their families.

## 1. Introduction

Autism spectrum disorder (ASD) is a complex neurodevelopmental condition that significantly affects social communication and is associated with restrictive interests and rigid or repetitive behaviors [1], with the degree of functional impairment differing significantly between individuals [2]. The phenotype of autism, referring to observable traits or characteristics, is manifested by an expansive range of cognitive and behavioral features which are defined as abnormal in quality or intensity when compared to typically developing peers [3]. The etiology of ASD involves a complex interplay between inheritance and environmental factors influenced by epigenetics. Over 800 genes and dozens of genetic syndromes are associated with ASD [4,5].

The term “syndromic autism” refers to ASD associated with clinically defined genetic syndromes that can be traced to the deletions or duplications of specific genes, with the autism phenotype manifested in association with neurodevelopmental delays. The genetic mutations or defects in single genes associated with syndromic autism result in a significant elevation of risk for developing ASD compared to the population risk, which is currently estimated at approximately 2.8%, with boys affected about four times more than girls [6]. Syndromic autism represents approximately 10% of all ASD cases and is often associated with congenital malformations or characteristic dysmorphic features [7]. In contrast to “idiopathic” or “primary” autism with increased incidence in males, syndromic autism has an equal male to female sex ratio [8].

There have been significant discoveries in relation to syndromic autism which have expanded our knowledge and understanding of factors underlying the etiology and development of these disorders [9]. The identification of these disorders depends on the ability of the clinician to recognize the clinically observable and identifiable features of the specific syndrome, with the diagnosis typically confirmed by targeted genetic testing (e.g., the mutation and repeat expansion screening of the FMR1 gene causing fragile X syndrome) [10,11]. A high index of suspicion for syndromic autism is warranted given that a growing number of individuals with ASD (now greater than 50%) are being diagnosed with genetic defects, syndromes, or chromosomal abnormalities. Advances of genomic technology with next-generation sequencing and high-resolution chromosome microarrays for the identification of small DNA deletions or duplications, gene variants, single gene conditions, or chromosomal defects as well as laboratory testing for metabolic disturbances and mitochondrial dysfunction can provide etiological evidence for neurodevelopmental disorders including ASD [12,13,14,15]. 

Of particular interest when evaluating for behavioral or psychiatric disorders in syndromic autism is the recognition of the autistic phenotype as a part of the broader neurodevelopmental presentation, with the identification of behavioral phenotypes being a key element in the field of clinical genetics [16]. Behavioral phenotypes, such as observed in syndromic autism, are defined by characteristic differences in cognitive, social, communication, or motor skills which are consistently demonstrated by individuals with the identified genetic findings [17]. From a diagnostic standpoint, the identification of the behavioral phenotype is as important as recognizing dysmorphic features and should include assessments of cognitive and intellectual functioning, reciprocal social and communication skills, attentional ability, and behavioral disturbances [18].

In the diagnostic evaluation of ASD, co-occurring conditions which can have overlapping symptoms or features including other developmental, behavioral, or psychiatric disorders must be taken into consideration [5]. Individuals with autism have an increased risk for neuropsychiatric disorders compared to the general population [19]; the odds of which are further elevated with syndromic autism [20] and even more so when associated with intellectual developmental disability (IDD). In cases with a greater severity of IDD, the diagnosis of ASD becomes increasingly difficult given their shared phenotypic features, making it essential to establish whether the observed ASD characteristics are not simply a manifestation of the global developmental delay [21].

The aim of this review is to summarize the specific characteristics, general clinical presentation, genetic correlates, and commonly occurring neurodevelopmental, behavioral, and psychiatric features of 12 selected classical genetic syndromes frequently associated with ASD, referred to as syndromic autism. Table 1 shows a list of these classic genetic syndromes associated with ASD and other neuropsychological, behavioral, and psychiatric findings with the involved gene(s) or chromosome region, the estimated prevalence of the genetic syndrome in the general population, and the estimated rate of ASD associated with each syndrome. It should be noted that a discussion of specific therapeutic approaches or interventions available for the treatment of behavioral and psychiatric disorders associated with syndromic autism is beyond the scope of this review.

## 2. Classical Genetic Syndromes

Genetic Syndromes Frequently Associated with Autism Spectrum Disorder

Fragile X SyndromeTuberous Sclerosis ComplexPhelan–McDermid SyndromePrader–Willi SyndromeAngelman SyndromeDiGeorge SyndromePhenylketonuriaDown SyndromeRett SyndromeWilliams SyndromeBurnside–Butler SyndromeCornelia de Lange Syndrome

### 2.1. Fragile X Syndrome

Genetic features:

Fragile X syndrome (FXS) is recognized as the second most common cause of intellectual disability (after Down syndrome), accounting for about 50% of X-linked intellectual disability, occurring in about 1 in 4000 male and 1 in 8000 female births [22]. FXS occurs due to a defect of the FMR1 (Fragile X Messenger Ribonucleoprotein 1) gene, due to a trinucleotide (CGG) repeat with more than 200 repeats in the FMR1 gene in the full mutation state [23]. This CGG triplet expansion produces instability in the FMR1 gene, leading to the abnormal methylation and suppression of transcription, resulting in decreased brain protein levels encoded by the gene required for normal neurodevelopment and function [24,25].

The premutation occurs when the repeats are in the range of 55 to 200 and can result in either early menopause (fragile X-associated primary ovarian insufficiency), fragile X-associated tremor ataxia syndrome (FXTAS, which is a progressive neurodegenerative disorder occurring later in life), or fragile X-associated neuropsychiatric disorder (FXAND). It should be noted that lesser degrees of functional impairment typically noted in females with FXS are due to the protective factor of having one normal FMR1 allele on their second X chromosome [25,26].

Neurodevelopmental aspects:

Autism is seen in approximately 50% of males and 20% of females with FXS meeting the full diagnostic criteria for ASD; however the ASD features can be relatively mild, particularly in females [27]. Among the genetic causes of autism, FXS accounts for up to 6% of all cases, with the resulting consensus that all individuals diagnosed with ASD should have fragile X DNA testing. Even in those not meeting the criteria for ASD, milder ASD-like features including social anxiety, extreme shyness, and eye gaze avoidance are common in FXS [28].

Fragile X syndrome is characterized by a moderate-to-severe intellectual disability, accompanied by distinctive facial features with a long narrow face and protruding jaw, large ears, mitral valve prolapse, joint laxity, soft skin, obesity, hernias, and macro-orchidism (enlarged testicles in 80 percent of males) [10]. Intellectual developmental disability affects approximately 85% of males and 25% of females with FXS, with intelligence quotients (IQ) ranging from an average to severe intellectual disability, noting that a range of neurocognitive strengths and challenges are observed in individuals with FXS. There is a general trajectory of decline in IQ scores with increasing age in males with FXS [29]. Females can have learning impairments, but these are generally not as severe as in males, social anxiety, early menopause, and increased odds for pregnancies with fraternal (dizygotic) twins [30].

Neuropsychiatric disorders:

FXS is associated with elevated rates of attention deficit hyperactivity disorder (ADHD), with estimates upwards of 90% causing many to regard ADHD-related deficits as part of the core FXS behavioral phenotype [31]. Up to 70% of young people with FXS have at least one anxiety disorder compared to about 10% of age-matched individuals in the general population. With onset typically in childhood, generalized anxiety disorder, specific phobias, social anxiety disorder, or obsessive–compulsive disorder are the most common psychiatric disorders associated with FXS [32]. Depression is also common in FXS, occurring in approximately 40% of premutation carriers and 65% of Fragile X-associated tremor ataxia syndrome, in which depressive symptoms typically manifest prior to the onset of motor symptoms, suggesting that depressive symptoms could represent a prodrome of later motor impairments in patients who develop FXTAS. Studies of depression in carriers found a significant predictive relationship between the repeat size and depression, demonstrating that repeats above 100 were associated with significantly higher ratings of depression severity [33].

### 2.2. Tuberous Sclerosis Complex

Genetic features:

Tuberous sclerosis complex (TSC) is an autosomal dominant multisystem genetic disorder; however, 70% of cases result from de novo (sporadic) germline mutations. TSC is classified as a neurocutaneous syndrome, or a phakomatosis (birth mark) disorder, with an incidence of about 1 in 6000 births, associated with two separate genes located on either chromosome 9 (TSC1) or 16 (TSC2). About 10 to 30% of cases of TSC are due to mutations of the TSC1 gene; however, in sporadic TSC, there is an even greater excess of mutations in TSC2, with TSC2 mutations associated with more severe disease [34].

The protein product of the TSC1 gene is known as ‘hamartin’ [23]. The loss of the tumor suppressor gene TSC1 is responsible for hamartoma-type (defined as abnormal growths consisting of the same tissue from which it derives) benign tumors originating within the kidneys, liver, heart, eyes, lungs, skin, and central nervous system. These tumors, although non-malignant, have the potential to grow and damage the affected organ system [35]. The severity of TSC between individuals can vary widely, in part determined by the number, size, and location of these tumors. Pulmonary tumors can include lymphangioleiomyomatosis (a rare lung disease characterized by abnormal cell growth in the smooth muscle tissue of the lungs and the abdomen), while skin lesions include melanotic macules, facial angiofibromas, and patches of connective tissue nevi or light-colored birthmarks. Kidney changes can include angiomyolipomas, renal cysts, hemorrhage, renal cell carcinomas, and the replacement of healthy renal tissue which may lead to renal failure [36]. Changes in the CNS associated with tuberous sclerosis complex (including cortical tubors and subependymal nodules or giant cell astrocytomas) can cause seizures, intellectual disability, learning or cognitive impairments, and psychiatric disorders [37].

Neurodevelopmental aspects:

The estimate of autism prevalence in TSC varies depending on the studies, with an incidence ranging from 40% to 50%, making it a leading genetic cause of syndromic autism. Numerous factors have been identified as being associated with ASD in TSC, including the brain lesion load, prominent lesion type, size and location of the tubers, presence of cyst-like tubers, TSC2 mutation, early onset and treatment refractory seizures, and the presence and severity of cognitive impairment [38].

There are a wide range of presentations and findings, as is often the case with autosomal dominant genetic disorders [39]. Seizures are one of the most common neurological symptoms in TSC, affecting about 85%, often presenting in infancy. Intelligence in individuals with TSC is in the normal range for about 44%, with intellectual developmental disability (IDD) impacting the remainder, with levels of IDD ranging from mild in about 28%, moderate in about 15%, and severe or profound in about 12%, with a greater likelihood of IDD with TSC2 mutations. Learning disorders in mathematics, reading, writing, or spelling have an incidence in TSC of nearly 60%. Some children with TSC appear to develop cognitively at a normal rate until the onset of seizures, at which point their acquisition of skills slows or regresses [40].

Neuropsychiatric disorders:

Most individuals with a mutation in the TSC1 or TSC2 genes develop neuropsychiatric symptoms during their lifetime, with TSC2 mutations associated with a significantly greater risk compared to TSC1 mutations. Emotional and behavioral problems are common, impacting about 50% of children with TSC, with a significant increase in frequency for those with intellectual disability [41]. Behavioral presentations in TSC often include severe aggressive outbursts, recurrent temper tantrums, ADHD-type symptoms including hyperactivity and impulsivity, speech and language delays, reduced eye contact, poor social skills, impaired relationships with peers, obsessions, repetitive behaviors, self-injurious behaviors, eating problems, alcohol dependence, or insomnia [42].

Anxiety and depressive disorders impact as many as two-thirds of people with TSC, often with onset in adolescence or adulthood [43]. It should be noted that mood and anxiety symptoms in those with neurodevelopmental disabilities often manifest differently than in typically developing individuals. Depressive-like symptoms including social isolation, decreased interest in previous activities, loss of appetite, or dysregulated behaviors can occur due to underlying physical health problems such as seizures, renal failure, or chronic pain, or treatments including anticonvulsant medication [44].

### 2.3. Phelan–McDermid Syndrome

Genetic features:

Phelan–McDermid syndrome (PMS) is one of the most common single-gene disorders associated with autism, with an occurrence rate of 2.5–10 per million births. PMS is caused by a sporadic deletion of the chromosome 22q13.33 band within a contiguous gene deletion region located distally in the long arm of chromosome 22, or by a mutation in the SHANK3 gene within the minimal critical chromosome region. SHANK3 is a gene which encodes a scaffolding protein that helps organize the glutamatergic postsynaptic density, thus playing a crucial role in the development of synaptic functions in the brain. PMS is one of the top 10 disorders found in the genetic screening of individuals with learning deficits and autism [13].

The SHANK3 gene is important for speech and language development as well as social communication and relationships with overall physical and neurological development. Those with only an abnormal dosage of the SHANK3 gene usually have severe cognitive deficits, including severe language and speech disorders, as well as many features typically associated with autism spectrum disorder [23]. It is noted that the overlap of features between PMS and other genetic causes of neurodevelopmental disorders makes a definitive clinical diagnosis without cytogenetic confirmation impossible, thus leading to the underdiagnosis of PMS [45].

Neurodevelopmental aspects:

Autism has been reported to affect upwards of 90% of individuals with PMS, in which the presentation of typical ASD features increases with age [46]. Comparative studies suggest that the manifestation of specific autistic characteristics in PMS may be different compared to that of most individuals with ASD, who generally demonstrate less social withdrawal, a greater frequency and severity of mood symptoms, and fewer repetitive behaviors [47].

PMS is typically characterized by neonatal hypotonia with a global developmental delay, the late development of motor skills, absent to severely impaired speech, feeding difficulties, inappropriate chewing behavior, and dysmorphic features of the face and hands including facial asymmetry, a prominent brow, small chin, prominent ears, ptosis with epicanthal folds, bulbous nasal tip, large hands, and dysplastic nails [23]. Also often seen with PMS are structural brain abnormalities, accelerated somatic growth patterns, cardiac abnormalities, gastrointestinal problems, renal malformations, sleep disturbance, a diminished perception of pain, and a decreased perspiration ability that results in proneness to hyperthermia [48].

The PMS phenotype is highly variable with the severity of developmental delays varying with the deletion size. Individuals with the same size deletion may be vastly different in their degree of limitations, but a moderate-to-profound intellectual disability is typical and seizure disorders are common [49]. Most children with PMS acquire basic motor skills such as walking, with gross motor functioning stronger than fine motor. Language delays with compromised expressive language skills are common, as are periods of regression in social and communication skills, or imaginative play, typically in early childhood. In PMS, the global developmental delays become more striking with maturation, a phenomenon known as ‘growing into deficit’ [50].

Neuropsychiatric disorders:

Behavioral problems associated with PMS include the frequent mouthing of objects, pica (eating non-food items), impulsivity, poor social skills, stereotypies, aggressive behaviors, recurrent self-injury, and repetitive self-stimulatory behaviors. Although behavioral difficulties generally decrease with age, in adults, difficulties in the areas of social relationships and anxiety often become more pronounced. Regression is defined as a prolonged loss of skills previously acquired and can occur during or following psychiatric episodes or other stressors such as infections, mood episodes, major life changes, or environmental stress. Loss of speech is most frequently reported, but loss of motor, communicative, and social interaction skills have been described at variable ages beyond the first three years of life, with many of those who experienced regression able to regain skills months or even years later [51].

Psychiatric disorders in PMS include anxiety disorders, obsessive–compulsive disorder, depressive disorders, bipolar mood disorder, and schizophrenia spectrum disorders. Onset can occur in mid-to-late childhood; however, most cases onset during adolescence or in the young adult years. Regression in cognitive skills, behavioral adaptation, or neurological functioning are reported in many with PMS following the onset of psychiatric symptoms [52]. Mood symptoms including depression or manic episodes are the most common and can be accompanied by anxiety and irritability. Episodes of disorganized behavior or psychosis are also common, sometimes occurring as a single episode, but often as recurrent episodes, with episode lengths varying between days and months. The risk of catatonia, a unique syndrome of motor and autonomic dysregulation often associated with a variety of psychiatric and medical conditions, has been reported to be elevated in PMS [53].

The treatment of behavioral and psychiatric disorders in PMS is complicated by unique pharmacogenomic considerations [54]. Given that the cytochrome P450 2D6 enzyme, which metabolizes 20% of approved drugs, including many prescribed for psychiatric conditions, is encoded by the CYP2D6 gene, which maps to 22q13.2 and is lost in PMS patients, with deletions larger than 8 Mb. There are many factors that contribute to drug response; however, given that there are over 100 known variants of CYP2D6, pharmacogenomic testing can be used as a tool in personalized care, with the potential to improve health outcomes and minimize the risk for adverse drug reactions [55].

### 2.4. Prader–Willi Syndrome

Genetic features:

Prader–Willi syndrome (PWS) is the most common syndromic cause of obesity and the most common genetic etiology of life-threatening obesity, with an occurrence rate of between 1/20,000–1/30,000 births [56], affecting about 500,000 affected individuals worldwide [57,58,59]. PWS is a contiguous gene syndrome due to abnormal DNA methylation in the Prader–Willi critical region (PWCR) within chromosome 15q11-q13. PWS is caused by errors of genomic imprinting, most often due to a paternal deletion of the chromosome 15q11-q13 region (the DEL subtype) in 60% of cases, or maternal uniparental disomy 15 (the mUPD subclass) in about 35% of cases. There are two deletion types in PWS, the larger Type I and smaller Type II. The remaining affected individuals have either a defect in the imprinting center controlling the imprinted genes in the PWCR or chromosome 15 translocations or inversions [60,61,62]. The diagnosis and molecular cause of PWS can be identified using high-resolution chromosome microarrays [61] or, more precisely, with a streamlined molecular approach including whole-exome sequencing with a deletion or duplication status, and DNA methylation analysis with methylation-sensitive multiplex ligation-dependent probe amplification (MS—MLPA) [63].

Characteristic features in PWS including serious health concerns include compulsive skin picking, strabismus, scoliosis, osteoporosis, hypogonadism (genital hypoplasia), small hands and feet, short stature, low level of growth hormone, and other hormonal deficiencies including hypothyroidism and central adrenal insufficiency [62,64]. The progression of distinct nutritional phases in PWS has been described as a consistent feature across PWS molecular genetic classes [57,59]. PWS is characterized by decreased fetal activity, infantile hypotonia, poor suck with feeding difficulties, and low body weight in infancy with early failure to thrive, which later progresses to hyperphagia (excessive hunger) in mid-childhood with the subsequent onset of morbid obesity, unless food intake is strictly limited [65,66,67,68]. Obesity in PWS represents one of the most serious health problems caused by hyperphagia and compounded by a decreased metabolism, lethargy, poor muscle tone, physical inactivity, and an inability to vomit [60,68].

There is a significant risk of premature death in PWS starting in childhood, with annual PWS-related mortality being approximately three times higher than the overall US population, historically with an expected average lifespan of just over 30 years. The leading causes of death in PWS are respiratory-related, accounting for more than half of all deaths, and include aspiration, lung infections, sleep-disordered breathing, and respiratory failure, followed by cardiac and gastrointestinal problems, and other infections including sepsis [69,70]. Growth hormone (GH) treatment along with lifestyle interventions, including strict adherence to a restricted caloric diet and routine physical exercise, can be an effective approach to prevent the development of obesity in PWS, with the potential to increase life expectancy [71,72,73].

Neurodevelopmental aspects:

ASD is typically diagnosed in about 26% of individuals with PWS, with the mUPD genetic subtype having a significantly increased risk of co-occurring ASD [74]; however, clinically impairing ASD symptoms occur in both the mUPD and DEL genetic subtypes [75,76]. However, it should also be noted that PWS is associated with a distinct behavioral phenotype which, among other things, includes fixated interests (in this case, regarding food specifically), obsessive–compulsive tendencies, excessive anxiety, rigid behaviors, temper outbursts, and deficits in social cognition, which in many respects significantly overlap with characteristic features of ASD [77,78].

Mild-to-moderate IDD with an average IQ between 60 and 70, frequently associated with attentional deficits, cognitive rigidity, and deficiencies in basic language skills are typical in PWS [79]. IQ is impacted by the PWS molecular genetic classes, as those with mUPD compared with DEL have, on average, a higher verbal IQ and greater numerical calculation skills, attention, word meaning, factual knowledge, and social reasoning, often resulting in a delay in PWS diagnosis [67,80]. In the absence of GH treatment, children with PWS tend to experience a deterioration in abstract verbal reasoning and visuospatial abilities over time. Growth hormone treatment prevents this deterioration, with an overall improvement in the total IQ demonstrated after four years of GH administration [81] and confirmed after eight years of GH treatment to the extent that up to 30% of patients with IDD eventually test in the normal range of intelligence [82]. Other studies examining the effects of GH treatment on cognition in PWS also showed a positive impact, supporting the role of GH on not only body composition and stature, but also intelligence [67].

Neuropsychiatric disorders:

PWS is associated with a wide spectrum of maladaptive and problematic behaviors. Typical behavioral features associated with PWS include hyperphagia-driven behaviors (e.g., food obsessions, hoarding, or foraging), temper tantrums, aggression, self-injurious behaviors, and stubborn, rigid, inflexible, repetitive, ritualistic, or manipulative behaviors. There is often insistence on sameness with demands for strict adherence to routines. Compulsive behaviors in PWS emerge as early as one year of age, with a mean onset at age five. They typically occur at a level which interferes with adaptive functioning, with the potential to lead to serious health and general safety concerns, causing significant stress for the individual as well as for their families, caregivers, and support network [83]. Those with the larger Type I deletion have more behavioral and psychological problems, poorer adaptive behaviors, and lower academic achievement than those with the smaller Type II deletion or mUPD [84].

Nearly 90% of patients with PWS over 12 years of age have at least one psychiatric disorder [85]. Common psychiatric symptoms include excessive anxiety with an average onset at age eight, which is a persistent issue across most life stages for individuals with PWS. Even though the emergence of anxiety typically occurs around the same age when hyperphagia develops, the evidence suggests that anxiety typically observed in PWS is not solely or primarily food-related [86]. Studies have found that rates of significant mood lability and affective mood disorders including major depression and bipolar disorder in PWS are increased in the mUPD subtype [68]. Most notable, however, is that the mUPD genetic subtype also has a strikingly higher prevalence of psychosis, including a phenomenon referred to as “cycloid psychosis” [87], with an estimated lifetime prevalence for psychosis of greater than 60%, compared with the deletion subtype that has a similar prevalence to adults with intellectual disabilities in the general population, which ranges from about 2.6 to 4.4% [88].

### 2.5. Angelman Syndrome

Genetic features:

Angelman syndrome (AS), first described in 1965, was the second discovered genetic disorder caused by errors in genomic imprinting, affecting approximately 1 in 15,000 births. AS involves the same deletion of the chromosome 15q11-q13 region, as seen in PWS, but of a maternal origin [89], whereas the 15q11-q13 region is paternally deleted in PWS [90]. There are four known genetic mechanisms that leads to AS, with about 70% resulting from a de novo maternal 15q11-q13 deletion, 2% from paternal uniparental disomy 15 (in which both copies of chromosome 15 come from the father), and 3% from an imprinting defect. The remaining 25% are caused by mutations in the UBE3A gene which encodes the enzyme ubiquitin protein ligase E3A, which regulates excitatory neural synapse development, a mechanism likely contributing to the cognitive dysfunction associated with AS and possibly other syndromic autisms as well [91,92].

Angelman syndrome is a neurodevelopmental disorder characterized by severe intellectual disabilities, with global developmental delays first noted between three and six months of age, and the unique clinical features of AS emerging generally after one year of age. Dysmorphic craniofacial features in AS include microcephaly, a large mandible with an open-mouth expression and protruding tongue, with other common features including impaired or absent speech, movement or balance problems, psychomotor retardation, ataxia with a wide-spaced gait, tremors, and jerky limb movements [93].

Neurodevelopmental aspects:

The prevalence of ASD phenomenology in AS is about 34% [94]. Phenotypic presentations in Angelman syndrome are highly dependent on the specific genetic subtype, with AS caused by the maternal deletion of 15q11-q13 most often associated with greater limitations in global development and more autistic features than those of other genotypes [95]. Seizure disorders onset by three years of age in 80% of patients, with akinetic seizures (characterized by an inability to initiate or maintain movements) demonstrated by characteristic electroencephalogram (EEG) patterns, often with epileptiform spikes even in the absence of observable seizure activity [96,97]. It is estimated that Angelman syndrome accounts for up to 6% of all children presenting with a severe intellectual disability and epilepsy [98].

AS has a unique behavioral phenotype that typically includes a persistently happy demeanor and sociable disposition, with frequent laughter and smiling, which has been regarded as a hallmark of Angelman syndrome since the original description of ‘happy puppet children’ [99]. Uncontrollable bouts of laughter can escalate into a nearly convulsive state and can occur unrelated to context [100]. Also common in AS are tongue thrusting, the frequent mouthing or chewing of objects, pica (swallowing non-food items), hyperphagia, restricted food preferences, and an affinity for water [101]. Stereotyped behaviors, compulsions, and rituals can include motor “stims”, which are typically purposeless repetitive movements, postures, gestures, or vocalizations. Motor stereotypies in AS generally manifest with fixed patterns, often focal (e.g., grimacing, bruxism, head shaking, finger wiggling) or whole-body (e.g., rocking, pacing, or jumping), with the most characteristic being hand flapping and waving. Repetitive and stereotyped behaviors have the potential for causing self-injury, such as in the case of compulsive eye-rubbing, which may result in damage to the cornea, with the potential for vision loss [102].

Neuropsychiatric disorders:

ADHD-type symptoms are manifested as part of the Angelman Syndrome behavioral phenotype, with typical social and behavioral aspects of AS demonstrated by an exuberant personality, impulsive tendencies, excitable behavior, and hyperactivity [97,103]. Distractibility with a reduced attention span is common, typically manifesting to a greater extent in AS compared to other neurodevelopmental disorders, and of a severity often sufficient to impair the recognition of potential hazards or dangerous situations [104,105]. Finally, it has been noted that stimulant medications, usually effective in most children for the treatment of ADHD, may be ineffective in managing “ADHD-type” behaviors in AS and are generally poorly tolerated in these children [106].

Anxiety is common in AS (despite a superficially happy demeanor), with up to about a fourth of children demonstrating clinically significant levels of anxiety. Although these children typically demonstrate an increased desire for social interaction, and usually have less fear of strangers or social anxiety compared to typically developing peers, specific phobias including fear of noise (phonophobia) and fear of crowds are common in children with AS, and can be triggered by changes in routine, unfamiliar surroundings, or separation from attachment figures. It has been shown that elevated anxiety in young people with AS is associated with higher levels of irritability, hyperactivity, social withdrawal, and disruptive behaviors [107].

### 2.6. DiGeorge Syndrome

Genetic features:

DiGeorge syndrome (DGS) is an autosomal dominant genetic disorder and is one of the most frequent microdeletion syndromes in humans, affecting about 1 out of 6000 births. DGS is caused by a chromosome 22q11.2 deletion of 1.5 to 3 Mb in size, with haploinsufficiency or loss of genes in the region including the TBX1 gene for heart development responsible for the phenotype (www.omim.org, accessed on 5 January 2024). This deletion occurs mainly as a de novo event; however, familial inheritance is noted in about 15% of cases [108]. Disorders due to the 22q11.2 deletion, which have overlapping features with DiGeorge and are collectively referred to as 22q11.2 DS (deletion syndrome), include Shprintzen, velo-cardio-facial, conotruncal anomaly face syndrome, and the CATCH 22 syndrome (acronym referring to cardiac defects, abnormal facial features, thymic hypoplasia, cleft palate, and hypocalcemia) [109].

The phenotype of DGS is caused by an embryonic disturbance of the cervical neural crest migration into the third and fourth pharyngeal arches and pouches leading to congenital anomalies, including parathyroid and thymus hypoplasia resulting in hypocalcemia and structural cardiac defects [23]. Hypoplasia of the thymus gland leads to immunodeficiency due to a deficit of T cells, while neonatal hypocalcemia causes tetany (recurrent muscle spasms) and seizures. Other common health issues in DGS include ophthalmological disorders, hearing deficits, dental abnormalities, scoliosis, renal disease, and dermatologic problems. Typical dysmorphic features include a short stature, telecanthus, abnormal slanting palpebral fissures, micrognathia, a bulbous nose, short philtrum, small mouth, hyper nasal voice, and palatal anomalies including a cleft palate [110]. Brain imaging findings in DGS include cortical malformations consisting of perisylvian polymicrogyria, most commonly in the right hemisphere [111].

Neurodevelopmental aspects:

The ASD diagnostic criteria can be met by up to 50 percent of children with DGS [112]. Neuroanatomical correlates between the two disorders are remarkable, as individuals in both groups display a reduced cerebellar volume and increased amygdala volume in neuroimaging studies. The amygdala is thought to play a role in quickly evaluating the emotional valence of incoming information and, being early in the emotional processing circuit, is well positioned to evaluate information downstream at the cortical level [113]. This is also consistent with research in autism documenting amygdala alterations, with the “amygdala theory of autism” positing that these findings are etiologically relevant to the autism phenotype [114].

Intelligence in DGS most often ranges from normal to mild intellectual disability, with the average Full-Scale IQ reported to be around 75, with learning disorders, especially in math and reading comprehension, very common [115]. Developmental and educational concerns are frequently reported in DGS, with impaired gross and fine motor skills, significant delays in language onset, speech problems, and expressive language difficulties. The variability in cognitive performance in DGS is, in part, related to the mode of inheritance of the deletion, with familial deletions resulting in a more severely affected phenotype compared to de novo deletions [116].

Neurological manifestations associated with DGS include seizures and Parkinson’s disease [117]. Seizure disorders occur with increased frequency in DGS, can emerge at any age, can be provoked or unprovoked, and are of either a generalized or focal onset. About 4% of adults with DGS have epilepsy, which is significantly higher compared to the general population risk of approximately 0.5–1% [118]. Movement disorders, including dystonias, non-epileptic multifocal and generalized myoclonus (sudden, brief involuntary twitching or jerking of a muscle or group of muscles), as well as early-onset Parkinson’s disease (PD) can occur as part of the expression of 22q11.2DS, with DGS responsible for approximately 0.5% of early-onset PD [119].

Neuropsychiatric disorders:

Neurodevelopmental and neuropsychiatric disorders associated with DGS impact more than half of the affected individuals [120]. The behavioral phenotype of children with 22q11.2DS has been described as overactive, impulsive, emotionally labile, shy, withdrawn, aggressive, or disinhibited [121]. Relative to unaffected siblings and typically developing controls, children with 22q11.2DS are described as having a more difficult temperament, being generally less cheerful in their usual demeanor, less able to maintain attention to a task, less regular in their daily habits, more rigid and inflexible, and having more difficulty responding and adapting to changes in the environment or routines [122].

ADHD prevalence in DGS has been reported to be around 40%, nearly five times higher than in the general population, and more frequently with the inattentive ADHD subtype in comparison to the combined hyperactive–impulsive subtype which is more common in idiopathic ADHD [123]. Although the elevated risk for ADHD in DGS has long been recognized, little is known about the pathogenesis explaining the association. ADHD is associated with the weaker function of prefrontal cortex (PFC) circuits, which play a crucial role in regulating executive function skills [124]. In DGS, the hemizygous 22q11 deletion encompasses the COMT (catechol O-methyltransferase) gene which codes for the enzyme COMT, responsible for degrading dopamine and norepinephrine, monoamine neurotransmitters essential for synaptic signaling in the PFC [125].

Anxiety disorders impact about a third of individuals with DGS, with the most frequent types including generalized anxiety disorder, separation anxiety disorder, social anxiety, panic disorder, specific phobias, and obsessive–compulsive disorder [126]. Remarkably, of individuals with DGS who have anxiety disorders, about two-thirds have more than one anxiety disorder. Anxiety disorders in DGS may emerge either in childhood or adulthood, but are more frequent in children and adolescents than in affected adults, and are more common in females as well as those with cognitive impairment [127].

A four-fold risk for developing psychotic disorders in adolescents and adults with DGS was first reported in the early 1990s, with a growing body of evidence that has emerged since [128]. DGS is considered one of the strongest known risk factors for schizophrenia and other psychotic disorders and is estimated to be etiologically responsible for up to 2% of all cases of schizophrenia [127]. The genetic risk for schizophrenia in DGS is related to deletion and in the absence of a family history of psychotic mental illness [129]. The risk of developing schizophrenia in 22q11.2DS is associated with the loss of several genes involved in the mitochondrial physiology, with changes in the neuronal mitochondrial function being caused by the haploinsufficiency of mitochondria-associated genes within the 22q11.2 region, including PRODH, MRPL40, TANGO2, ZDHHC8, SLC25A1, TXNRD2, UFD1, and DGCR8 [130].

Psychotic disorders affect approximately 10% of adolescents with DGS, with the overall prevalence of psychosis increasing to about 30% in adults with DGS, with the more specific psychiatric diagnosis of schizophrenia or schizoaffective disorder increasing with age. The only groups at higher risk for developing schizophrenia are individuals with two parents having schizophrenia or monozygotic twins born to a parent with schizophrenia [131]. The manifestation of schizophrenia associated with DGS is indistinguishable from other forms of schizophrenia with respect to their prodromal features, age at onset, core symptoms, and cognitive profile, except for the overall lower average cognitive and intellectual functioning and an absence of sex differences [132]. Early cognitive decline in DGS has been shown to be a robust indicator of the risk for developing psychosis, a prodrome like that observed in idiopathic schizophrenia [133]. Remarkably, the risk of schizophrenia is elevated six-fold in those with clinically significant levels of anxiety. Early intervention in the subgroup of children with subthreshold signs of psychosis and internalizing symptoms, especially anxiety symptoms, may reduce the risk of developing psychotic disorders during adolescence [134].

### 2.7. Phenylketonuria

Genetic features:

Phenylketonuria (PKU) is an autosomal recessive metabolic disorder, one of the first recognized inborn errors of a metabolism defect to be identified, and the most common genetic metabolic disease, affecting about 1 in 12,000 births. PKU is caused by mutations in the phenylalanine hydroxylase gene encoding the enzyme phenylalanine hydroxylase which breaks down the amino acid phenylalanine (Phe), with this enzyme accounting for about 75% of the disposal of the phenylalanine ingested from dietary sources [135].

It is important to diagnose PKU at a very young age because if undiagnosed or untreated, PKU has significant negative impacts on cognition and postnatal development, since high levels of Phe are toxic to the developing brain. As it can be detected by a simple blood test and is treatable, PKU has been included in newborn state screening programs nationally, with treatment consisting of a low phenylalanine diet starting at birth and continuing for life. Unfortunately, the late diagnosis of PKU still occurs in countries with no newborn screening programs or without national coverage [136].

Early diagnosis and compliance with a strict diet with the maintenance of low blood Phe levels will help maintain long-term health for those with PKU, but even with dietary control, phenylalanine levels in treated individuals can remain elevated. Persistent elevations of Phe can cause adverse neurocognitive and neuropsychiatric outcomes. Untreated PKU results in the accumulation of Phe in the body, leading to significant neurodevelopment sequelae including intellectual disability, growth retardation, and seizures, as well as a range of neuropsychiatric sequelae [137].

In the case of a mother with untreated or poorly managed PKU during pregnancy, resulting fetal malformations and adverse neurodevelopmental outcomes can occur. Affected offspring are at risk for complications including congenital heart disease, craniofacial dysmorphologies, microcephaly, urologic abnormalities, intrauterine and postnatal growth retardation, intellectual impairments, learning disorders, behavioral problems, and developmental disorders including autism [138].

Neurodevelopmental aspects:

PKU has a well-documented association with ASD, although the prevalence seems to be relatively low, with a frequency ranging from 3 to 6 percent [139,140]. Early-treated PKU results in significantly improved neurocognitive outcomes, learning, and academic achievement; however, despite the significant neuroprotective effect of controlled dietary interventions, variable findings may occur. These include mild-to-moderate deficits in a range of domains, including general intelligence (with IQ scores mostly within the average range, yet, on average, lower compared to controls) as well as impaired attention, processing speed, working memory, learning and memory, fine and gross motor skills, motor coordination, and executive functioning (including inhibitory control, conceptual reasoning, planning, problem solving, organization, inhibitory control, and cognitive flexibility) [141].

Untreated PKU results in intellectual disability, unusual mousy body odor, lighter pigmentation (Phe is involved in skin and eye pigment production), postural changes, gait disturbances, eczema, and epilepsy. Prior meta-analyses have found a strong inverse relationship between blood Phe levels and intellectual functioning in pediatric populations [142,143]. Failure to implement treatment in the neonatal period causes neuropsychological impairments due to the toxic effects of excess Phe in the brain, resulting in a diffuse white matter pathology, compromising multiple CNS pathways, causing a broad array of clinical features [144]. Thus, untreated PKU is characterized by intellectual disability ranging in severity from mild to profound, microcephaly, autism, seizures, motor deficits, developmental delays, behavioral issues, and an array of psychiatric symptoms [145].

Neuropsychiatric disorders:

Disturbances in emotional, behavioral, and psychological functioning are common in PKU, caused by the detrimental effect of excess Phe on the CNS and resultant neurotransmitter deficiencies. Children with PKU have increased rates of irritability, hyperactivity, learning difficulties, and school problems and decreased motivation, social competence, and self-esteem [146]. Untreated PKU has long been known to be associated with significant neuropsychiatric disturbance, but even with early and continuous dietary management, there remains a significantly elevated risk of developmental and psychiatric impairments, the most common being ADHD, anxiety, depression, and psychosis [147].

ADHD occurs at about twice the rate in individuals with PKU compared with the general population, with a demonstrated dose-dependent relationship between phenylalanine levels and ADHD symptoms. Both PKU and ADHD are associated with low levels of dopamine, and there is significant evidence linking dopamine to the neurobiology of ADHD. The link between ADHD and PKU is consistent with a hypodopaminergic hypothesis, as dopamine deficiencies in the prefrontal cortex and striatum play key roles in the regulation of attention and impulse control [148].

Anxiety is one of the most frequent self-reported symptoms in PKU, with anxiety disorders including generalized anxiety, panic disorder, specific phobias, and obsessive–compulsive disorder being significantly more common in PKU populations compared to the general population [149]. Anxiety disorders have been associated with alterations in the serotonergic system, leading to low serotonin levels. Given that elevated levels of Phe in PKU result in a reduction of CNS serotonin, the increased prevalence of anxiety disorders in PKU is unsurprising. Anxiety symptom severity in early-treated children with PKU has been found to correlate with blood Phe levels. Patients who adhere to Phe-restricted diets into adolescence appear less likely to develop anxiety disorders, and those who develop anxiety disorders tend to have fewer symptoms once resuming adherence to dietary guidelines [150].

Depression rates are significantly elevated in PKU compared to the general population, with significant depressive symptoms reported by up to half of adult patients with PKU, with the risk for depression lower in early- versus late-treated PKU [151]. The brain is organized into distinct, functional neural networks and depressive disorders are increasingly recognized as diseases of network dysfunction. CNS monoaminergic networks projecting from the brainstem to the frontoparietal and corticolimbic regions are networks of neurons that utilize the monoamine neurotransmitters dopamine, norepinephrine, and serotonin, each of which play an essential role in mood regulation [152]. The elevated risk of depressive disorders in PKU may be explained by excess Phe which leads to the impaired functionality of neural networks and reduced monoaminergic transmission, supported by evidence from clinical trials demonstrating the correlation of acute elevations in blood Phe with associated declines in mood [153].

Psychotic disorders including schizophrenia spectrum disorder occur at about twice the expected rate in PKU than in the general population [149]. Phenylalanine is one of the amino acids utilized for the synthesis of dopamine, the neurotransmitter which plays a central role in the neurobiology of schizophrenia. The “dopamine hypothesis” of schizophrenia proposes excessive dopamine transmission in the CNS mesolimbic areas along with deficient dopamine transmission in the prefrontal cortex [154]. Additionally, neuroinflammation, implicated in the pathology of schizophrenia can impair the function of the enzyme phenylalanine hydroxylase (PAH), thus leading to the accumulation of Phe. Researchers measured plasma Phe in 950 patients with schizophrenia, with a comparable cohort of controls, finding elevated Phe levels, thus supporting the hypothesis of aberrant PAH functioning in schizophrenia [155].

### 2.8. Down Syndrome

Genetic features:

Down syndrome (DS) or trisomy 21 is the most frequent form of intellectual disability caused by a third copy of all or part of chromosome 21 (a microscopically recognized chromosome aberration) occurring in about 1 in 1000 pregnancies. DS is characterized by a well-defined phenotype with an increase in incidence correlated with an advanced maternal age. Trisomy 21 occurs either by the nondisjunction of chromosome 21, with the presence of 47 chromosomes, or by the translocation of an additional copy of chromosome 21 to another chromosome [23]. The development of cell-free noninvasive prenatal screening and the parallel sequencing of maternal plasma cell-free DNA has mostly replaced the use of invasive testing (i.e., amniocentesis) which carried with it significant risks, including pregnancy loss [156].

Major clinical findings seen in DS include congenital heart defects (often atrial ventricular septal defects), gastrointestinal tract anomalies (including duodenal stenosis or atresia, an imperforate anus, and Hirschsprung disease) and endocrine disorders. Typical dysmorphic features include a small chin, epicanthal folds, low muscle tone, flat nasal bridge, single crease of the palm, and protruding tongue which can lead to obstructive sleep apnea. Nearly half of individuals with DS have either speech, hearing, or vision disorders, and adults typically have a short stature and obesity. DS is associated with a 10 to 15 times elevation in the risk for malignancies, including testicular cancer and leukemia [157].

Neurodevelopmental aspects:

Autism spectrum disorder is common in persons with DS, with rates reported as high as 39% [158]. Studies comparing functioning between those with Down syndrome only (DS-only) versus DS plus ASD have found that individuals with DS + ASD typically have lower cognitive abilities when compared to those with DS-only, along with a greater impairment in social and communication skills and a higher degree of rigid and repetitive behaviors [159,160].

Intellectual disability in DS is most commonly in the moderate range, but can vary between mild and severe, whereas social function is often highly relative to cognitive impairment. The risk for epilepsy increases with age, with about 5 to 10% of children and up to 50% of adults affected. Adults with DS have a significant risk of early onset dementia, with about 15% of affected individuals 40 years or older developing Alzheimer disease [161].

Neuropsychiatric disorders:

The overall prevalence of maladaptive behavior in DS has been estimated at about 20% [162]. Externalizing behaviors such as defiance, aggression, and delinquency tend to decrease after puberty, while internalizing behaviors such as social inhibition and withdrawal have been reported to increase in adolescents and teenagers, along with an age-dependent increase in irritability, attention problems, and impulsivity [163].

Findings from a recent study involving the largest documented cohort (over 6000 individuals) with DS in the United States identified significant elevations in mental health conditions compared with age- and sex-matched controls. This retrospective study obtained 28 years of data and documented them compared to controls. Individuals with DS had a higher prevalence of mood disorders including depression, anxiety disorders including obsessive–compulsive disorder, personality disorders, conduct disorders, impulse-control disorders, and psychosis including schizophrenia. Also interesting was the finding that individuals with DS demonstrated an overall lower prevalence of ADHD, bipolar affective disorder, generalized anxiety disorder, panic disorder, specific phobias, post-traumatic stress disorder, and substance use disorders compared to that of the general population [164].

Adolescents and young adults with Down Syndrome, when compared to individuals with other causes of intellectual disability, had significantly higher rates of psychosis or depression with psychotic features (43% versus 13%). Psychotic symptoms, including hallucinations and delusional beliefs, are reported at surprisingly higher rates in females compared to males with DS, a finding in stark contrast to the lack of gender differences generally reported in psychosis, whether in samples from the general population or in groups of individuals with other causes of intellectual disability [165].

### 2.9. Rett Syndrome

Genetic features:

Rett syndrome (RS) is a complex neurodevelopmental disorder, occurring almost exclusively in females, associated with intellectual disability, communication deficits, and motor skill impairments, and affects approximately 1 in 10,000 female births. Rett syndrome is a neurodevelopmental disorder characterized by apparently normal early growth and development up to about 6 months of age, followed by regression which occurs typically between ages 12 and 24 months, but sometimes not until after 4 years. RS is a monogenic X-linked dominant genetic disorder related to a mutation in MECP2, which encodes the methyl-CpG-binding protein MeCP2, a transcriptional regulator involved in synaptic formation. Currently, about 900 MECP2 gene variants (benign and pathogenic) have been identified and different mutations may contribute to different levels of disease severity [166]. The clinical presentation of RS is highly variable, but general correlations with disease severity have been reported for specific genotypes [167].

The diagnosis of RS is based exclusively on a set of clinical criteria derived from an expert consensus. For the diagnosis of typical RS, the affected individual must have had a period of relatively normal development after birth, followed by acquired microcephaly, the regression of skills including volitional hand use and spoken language. Hand use is replaced by distinctive, purposeless, or unusual hand movements such as hand wringing, repetitive rubbing, squeezing, tapping, or clapping (stereotypies). The diagnostic criteria for RS include a period of regression with subsequent recovery or stabilization in addition to four characteristic neurologic features, including a loss of acquired hand skills, loss of spoken language, gait abnormalities, and hand stereotypies. Classic (or typical) RS manifests all four of the main criteria, while variant (or atypical) RS meets at least two of the four main criteria in addition to the additional supportive criteria. Lifespan expectations are reduced due to epilepsy-, respiratory-, and cardiac-related issues, as well as other health issues [168].

Neurodevelopmental aspects:

Autistic features such as social withdrawal and the avoidance of eye gaze occur often during the period of active regression, but may improve in the subsequent period of stabilization or recovery, which occurs typically in the early school years [169]. It has been reported that autistic features are present in approximately 50% of people with RS, but these features decrease with time so that, over time, many will no longer meet the criteria for ASD [170]. Interestingly, studies using computer-based eye-tracking devices indicate that individuals with RS have a preference to look at human faces, especially eyes, which contrasts with gaze aversion typical in autism [171]. An initial diagnosis of autism in individuals eventually diagnosed with RS [172] is more likely to occur in less severely affected individuals, particularly those with a milder atypical variant of RS referred to as the preserved speech variant [173].

RS has been historically associated with an assumption of uniform and severe intellectual impairment, an impression supported by the characteristic deceleration of head growth and a low brain weight. An accurate assessment of the cognitive skills of individuals with severe motor and communication limitations is challenging. Cognitive functioning in RS was traditionally assessed by cognitive tools that were heavily dependent on motor functioning. A novel test by which to assess receptive language abilities in RS uses eye-tracking technology (ETT), with remarkable differences in the receptive vocabulary demonstrated between individuals with RS using ETT, with a young age positively correlated with a higher receptive vocabulary. In a study of 17 girls (mean age six years) with RS, the verbal comprehension abilities of about one-third ranged from a low average to mild cognitive impairment, with the other two-thirds being in the moderate-to-severe range, thus demonstrating significantly higher cognitive functioning in RS compared to historical assumptions [174].

Rett syndrome is characterized by early neurological regression that affects motor, cognitive, and communication skills, along with other motor abnormalities including low muscle tone, ataxia, and apraxia, and often seizure disorder. Epilepsy affects up to 90% of individuals, with the most common seizure types including partial complex, tonic–clonic, and myoclonic seizures, usually starting after age four and generally diminishing in severity during adulthood [175]. Patients with RS are at a greater risk of sudden death, likely because epileptic seizures can potentiate brainstem vulnerability, thereby increasing the risk of sudden unexpected death in RS, especially in those with cardiac or respiratory disease. Specifically, underlying autonomic dysregulation in RS leads to fluctuations in the vagal tone, thus predisposing RS patients to sympathetic storming, with a subsequent deleterious effect on the ascending control of brainstem functions [176].

Neuropsychiatric disorders:

Behavioral problems are common in Rett syndrome and typically include both internalizing (e.g., social withdrawal, anxiety, depression, and mood lability) and externalizing (e.g., aggressive behaviors directed either towards self or others) components, although it should be noted that aggressive behaviors are generally limited in RS due to severe motor impairment. Self-injurious behaviors in RS can include head banging, biting, or chewing on fingers or hands, or the hitting of self, likely exacerbated by the common report of an increased pain tolerance. Outbursts of unexplained screaming or laughing are not uncommon in RS. The likelihood of either internalizing or externalizing behaviors in RS is linked to a younger age, better motor function, and mild MECP2 mutations, demonstrating that patients with a less severe clinical presentation with RS tend to have more prominent behavioral problems [177].

Anxiety symptoms commonly observed in RS include a fearful expression, generalized tension, inability to relax, nervousness, panic attacks, inconsolable crying episodes, tremulousness in the absence of fear-inducing stimuli, excessive worry, screaming outbursts, increased hand stereotypies, and the worsening of hyperventilation or breath-holding, especially when in a novel or stimulating environment [178]. Social anxiety appears to be particularly common in RS and often manifests with the avoidance of eye contact, excessive shyness, withdrawal from social contact, blunted emotional facial expressions, and difficulty initiating communication. Internalizing symptoms are generally more severe in individuals with mild MECP2 pathogenic variants when compared to participants with either moderate or severe pathogenic variants [179].

### 2.10. Williams Syndrome

Genetic features:

Williams syndrome (WS) is a classic multisystem disorder caused by a deletion of the chromosome 7q11.23 band containing approximately 30 genes, including the elastin (ELN) gene, responsible for the commonly associated cardiac and other features of WS, with an incidence of about 1 in 20,000 individuals [180]. Up to 95% of cases involve the loss of GTF2I and GTF2IRD1 genes, caused by nonallelic homologous recombination in a region of chromosome 7 containing blocks of low copy repeats with a high sequence homology that are predisposed to rearrangements during meiosis [181].

WS is characterized by a specific heart defect (supra valvular aortic stenosis) and distinctive facial features (sometimes referred to as ‘elfin-like’), with characteristic facial features including a broad forehead, medial eyebrow flare, periorbital fullness, strabismus, stellate iris pattern, flat nasal bridge, full cheeks and lips, smooth long philtrum, pointed chin, and wide mouth, with facial features usually becoming more course with advancing age [182]. Additionally, peripheral pulmonary arterial stenosis has been identified along with infantile hypercalcemia, connective tissue abnormalities, characteristic dental abnormalities, a hoarse, brassy voice secondary to vocal cord paralysis, and a short stature. Individuals with WS are often described as having a unique personality, variable cognitive profile, learning delays, and behavior problems including autism [183,184].

Neurodevelopmental aspects:

ASD and Williams Syndrome can be viewed superficially as opposites in the domain of social cognition and behavior. Individuals with ASD are normally characterized as socially withdrawn, possessing limited understanding of social norms, and having difficulty engaging in interpersonal interactions. This contrasts with the hypersocial behavior exhibited by those with WS, who are typically socially outgoing, overly friendly, and eager to interact with others. However, it should be recognized that both conditions are quite similar in many respects when social motivation is removed. Individuals with WS exhibit difficulties in the social arena that overlap with ASD, with about 20% of children with WS meeting the diagnostic criteria for ASD [185,186]. Parents of children with WS often report poor social skills, difficulties with understanding important social cues, and difficulty maintaining friendships, as well as behavioral traits of inflexibility, ritualism, compulsiveness, and obsessional tendencies [187]. Pragmatic language deficits (referring to difficulty communicating both verbally and nonverbally in social situations) are also common to both WS and ASD and can be hypothesized to derive from specific deficits that both groups have with inferring the mental states of others [188].

Cognitively, individuals with WS typically have mild-to-moderate ID or learning disabilities yet heightened expressive language. Relative to the overall level of intellectual ability, individuals with Williams syndrome typically show a clear strength in auditory rote memory and a strength in language skills, but an extreme weakness in visuospatial construction. The adaptive behavior profile for Williams syndrome involves clear strength in socialization skills (especially interpersonal skills related to initiating social interaction) and strength in communication, yet a clear weakness in daily living skills and motor skills relative to the overall level of adaptive behavior functioning [189]. People with WS exhibit a mean IQ of about 55 [180], with intellectual functioning remaining stable across adolescence and adulthood, but with adaptive functioning declining over time [190].

Neuropsychiatric disorders:

The personality of individuals with Williams syndrome is characterized by high sociability, empathy, and overfriendliness, but with an undercurrent of anxiety related to social situations. Parents of children with WS report significant levels of attention problems in addition to elevated rates of tantrums and physical aggression; however these concerns generally decline by the age of 18 [191]. Sensory integration is a term that has been used to describe processes in the brain that allow us to take information we receive from both internal sources (within our bodies) and external (from the environment), process it, organize it, and respond appropriately. Sensory integration difficulties are common to both WS and ASD. Children with WS who have poor sensory integration have been found to have more difficulty with emotional control, low frustration tolerance, emotional reactions, and problem behaviors [192].

Anxiety-related psychopathology is a seemingly paradoxical feature of WS, given a typically socially gregarious disposition; however, anxiety is the most significant mental health concern for Williams syndrome, with rates as high as 80% [193]. Remarkably, the rates of specific phobias, which occur in over 50% of individuals with WS, are higher than that of other anxiety disorders which occur often in WS, including separation anxiety, obsessive–compulsive disorder, and generalized anxiety disorder [194]. Interestingly, the content of phobias in WS are distinctive, with reported phobias often relating to noise stimuli and blood, injury, and injections, in contrast with common phobias in other individuals with ID, typically fears of ghosts and animals [195], which may be related to a heightened sensitivity to sounds (hyperacusis), and frequent hospitalizations due to health problems may lead to fears of blood and injury [196]. It has been shown that elevated levels of anxiety in WS tend to persist and often escalate over time, and that anxiety often contributes to impairments in social functioning, often resulting in subsequent social isolation despite a persistent strong desire for social interaction [197].

### 2.11. Burnside–Butler Syndrome

Genetic features:

The 15q11.2 BP1-BP2 deletion (Burnside–Butler) syndrome is an emerging disorder now recognized as the most common cytogenetic finding in patients presenting with neurodevelopmental or autism spectrum disorders [198,199]. Burnside–Butler syndrome is diagnosed using high-resolution chromosomal microarray genetic testing. Ho et al., in 2016 [13], reported a study of over 10,000 consecutive patients presenting with autism spectrum disorder or neurodevelopmental disturbances for high-resolution chromosomal microarray analysis. They found 85 different chromosome or genetic defects in their patient cohort with 9% having the 15q11.2 BP1-BP2 deletion followed by the 16p11.2 deletion, with six of the top ten neurodevelopmental disorders involving chromosomes 15 or 16 [13].

The four genes found in the 15q11.2 BP1-BP2 region are NIPA1, NIPA2, CYFIP1, and TUBGCP5. These genes are individually associated with Prader–Willi syndrome, fragile X syndrome, autism spectrum disorder, schizophrenia, epilepsy, and Down syndrome [200]. Specifically, NIPA1 and NIPA2 genes encode magnesium and other cation transporters which play a role in gait ataxia, neuromuscular function, and autism. CYFIP1, which plays an important role in neuronal cytoskeletal remodeling and has been associated with the abnormal white matter microstructure, is associated with microcephaly, autism, and fragile X syndrome [201,202]. Finally, TUBGCP5 (Tubulin Gamma Complex Component 5) is a protein-coding gene associated with chromosome segregation at the cell level [203].

Neurodevelopmental aspects:

From reported patient cohorts presented for genetic services and microarray analysis, this microdeletion syndrome can now be recognized as the most common cytogenetic abnormality found in ASD, with up to 50% of children with the 15q11.2 BP1-BP2 deletion meeting the criteria for a diagnosis of ASD [200,204]. Individuals with 15q11.2 BP1-BP2 deletion can present with a wide range of clinical findings; however, not all individuals with this microdeletion will present with identifiable clinical manifestations, with an active area of study ongoing to further explain the non-penetrance nature of this emerging microdeletion disorder [205].

Although one-third have cognitive abilities in the average range, global developmental delays with intellectual disability and language delays are found in more than two-thirds of individuals with 15q11.2 BP1-BP2 deletion. Additionally, there is an increased risk for microcephaly, brain imaging abnormalities, dysmorphic or congenital anomalies (most commonly affecting ears or palate), seizures, ataxia, balance issues, coordination problems, motor or speech delays, memory problems, and learning disorders including dyslexia, dysgraphia, and dyscalculia [199,206,207].

Neuropsychiatric disorders:

Burnside–Butler syndrome is associated with an increased risk for self-injurious behaviors, ADHD, oppositional defiant disorder, obsessive–compulsive disorder, and psychosis. Investigations of neurons derived from patients with the 15q11.2 BP1-BP2 deletion show abnormalities of dendritic spine formation [208], with evidence from genetic and molecular studies implicating structural alterations at spiny synapses in the pathogenesis of major neurological and psychiatric disorders [209]. The TUBGCP5 gene, contained within the 15q11.2 BP1-BP2 deletion, has been found to be specifically associated with ADHD and obsessive–compulsive disorder [203]. Deletions encompassing the BP1-2 region at 15q11.2 increase the schizophrenia and epilepsy risk, but only some carriers have either disorder. A reduced expression of the CYFIP1 gene, which is also impacted by this deletion, has been implicated in the dysregulation of schizophrenia-associated gene networks [201].

Copy number variants (CNVs, defined as structural mutations that occur when genomic regions are duplicated or deleted compared to the reference genome which can be identified by chromosome microarray analysis) such as the 15q11.2 BP1-BP2 deletion may uniquely impact clinical phenotypes beyond simply increasing the risk for schizophrenia, thus potentially serving as models for studying treatment resistance in major mental illnesses [210,211]. This is of significant clinical relevance as approximately one-third of individuals diagnosed with a psychotic disorder experience treatment-resistant psychosis [212]. A case report of a woman with treatment-resistant schizophrenia, who was found to have the 15q11.2 BP1-BP2 deletion, reminds clinicians to consider the possibility for rare CNVs which may help to explain atypical clinical presentations of severe mental illnesses which have an identifiable genetic origin, and thus help to guide potential treatment implications, available either now or in the future [213].

### 2.12. Cornelia de Lange Syndrome

Genetic features:

Cornelia de Lange syndrome (CdLS) is recognized as a multisystem malformation disorder with characteristic facial dysmorphism, pre- and post-natal growth retardation, learning impairment, behavioral problems, and upper limb anomalies. Estimated to occur in about 1 in 10,000 to 30,000 births, there is a wide range of clinical variability in this disorder, including milder phenotypes which are recognized [214]. CdLS is heterogenous and caused primarily by mutations of the NIPBL gene located in the chromosome 5p13.2 region in 50 to 60% of cases, with other genes causing this disorder, including SMC1A, SMC3, and RAD21. These four genes collectively encode components of the cohesin protein complex which mediates sister chromatid cohesin, essential for normal growth and development at the cellular level, including mechanisms required for normal DNA replication, thus impacting cell division and normal embryo development [215].

Cornelia de Lange syndrome is characterized by developmental aberrations in multiple organ systems. Characteristic facial findings in CdLS include a low anterior hairline, arched eyebrows, long eyelashes, synophrys, maxillary prognathism, long philtrum, thin lips, an unusually shaped small mouth, limb malformations, and digital defects [216], with other systems affected including the brain, bone, immune, endocrine, gastroesophageal, cardiac, ophthalmologic, genitourinary, and skin systems [217].

Neurodevelopmental aspects:

Autistic features, notably excessive repetitive behaviors and expressive language deficits, occur in CdLS with an incidence of up to 80% [218]. Between 65% and 85% of individuals with CdLS meet the criteria for ASD, but when compared to those with idiopathic ASD, individuals with CdLS are less likely to demonstrate repetitive and stereotyped behaviors and fewer sensory-related behaviors [219]. The nature of repetitive behaviors in CDLS appears to be different comparatively to other patients with ASD, with CdLS showing fewer sensory-related behaviors such as licking, sniffing, and finger flicking, and having relatively less affected social adaptive skills [220,221,222].

Intellectual developmental disabilities in CdLS may vary from borderline to profound. Verbal communication deficits are usually of a severity causing everyday self-help skills to be significantly limited [223]. Seizures are common, with partial epilepsy being the most common type of seizure disorder. Up to 45% of those with the SMC1A variant have epileptic seizures, although most have a favorable longer-term prognosis, in which antiepileptic medication can be withdrawn after a few years of complete seizure control [224,225].

Neuropsychiatric disorders:

A wide variety of behavioral symptoms reported in CdLS include hyperactivity, self-injury, daily aggression, and sleep disturbances, which correlate closely with the degree of learning impairment. Repetitive behaviors including ‘circling, twirling, whirling’ and hand posturing are universal, and recurrent self-injurious behaviors including head banging, hitting, and skin picking are common, increasing in frequency during adolescence, but often improving in early adulthood [219]. Adaptive behaviors in CdLS across the lifespan are typically impaired, more markedly in those with variants in NIPBL, with a tendency for daily living skills to decline with age [226].

Social anxiety, social avoidance, extreme shyness, and selective mutism are particularly common in CDLS, affecting about one-third. Anxiety symptoms often manifest as socially avoidant behaviors which can take the form of reduced verbal interaction or even selective mutism, as well as mounting anxiety when routines or ritualistic behaviors are not completed [227,228]. Behavioral or psychiatric issues including self-injury, autistic features, anxiety, attention deficits, depression, and obsessive–compulsive behavior often worsen with age, possibly pointing to abnormal aging in CdLS, a clinical phenomenon largely understudied to date [229].

## 3. Discussion

Hundreds of genes have been identified as playing a role in autism. Monogenic- or chromosomal-based disorders including syndromic autism can be relatively straightforward to diagnose with the recognition of syndromic features and the use of advanced genomic technology, such as next-generation sequencing at the exome level or disease-specific gene panels (e.g., autism), to identify single-gene defects or variants [4,230]. The role of the physician clinical geneticist is to evaluate the patient with features of autism, recognize the features, and then diagnose associated clinical genetic syndromes. This requires the use of appropriate information from the medical and family histories and the identification of dysmorphism related to specific genetic disorders prior to selecting the most appropriate genetic and laboratory testing to confirm the diagnosis, which might include a chromosomal microarray, the next-generation sequencing of single-gene or disease-specific-gene panels, or clinical or whole DNA exome analysis. Once testing results are available, then information will be shared with the patient and family. Genetic counseling is provided regarding the risk factors and disease-specific surveillance for the patient, with additional recommendations (e.g., head MRI, cardiovascular screening using echocardiograms, neuropsychiatric evaluations, and further laboratory assessments including metabolic or mitochondrial testing) based on the genetic defect or syndrome identified.

Pharmacogenomics is the study of DNA and RNA characteristics impacting gene function and drug metabolism, which is considered an emerging field in medical care and clinical practice impacting treatment. It involves the cytochrome P450 liver enzymes needed to metabolize drugs or medications, and the transporters, receptors, or sensitivity markers identified in the individual patient. The selection and length of treatment per patient may be unique, as may be their response to specific drugs, specifically behavioral and psychiatric medications [231]. Hence, evidence-based medication selection and dosing strategies may be driven by the pharmacogenetics or DNA pattern of liver enzymes and their normal, slow, or rapid metabolism status, unique to each individual patient. The use of this precision medicine approach in treating those affected with neurodevelopmental disorders including ASD and syndromic autism is important, as evidenced by the report from Forster et al. in 2021 when treating aberrant behavior in patients with Prader–Willi syndrome where the specific PWS genetic subtypes and defects may impact selection and treatment options for medication use [232]. Depending on the clinical presentation, other diagnostic genetic laboratory tests may be considered, including gene repeat expansion analysis, e.g., FMR1 gene for fragile X syndrome [25], high-resolution chromosome microarray studies [13], or other laboratory methods including mitochondrial testing [12]. Laboratory-based methods are generally excellent in identifying genetic defects associated with specific identifiable biological markers that can be evaluated and followed, with consideration of potential treatment avenues available or discovered over time.

Limitations of this review include a lack of discussion of the in-depth analysis of the underlying mechanisms involved in syndromic autism, as well as theoretical approaches that might explain the links between specific genetic mutations or gene–gene or protein interactions that may impact behavioral manifestations of syndromic autism and their treatment. Similarly, our report did not discuss technological advances such as next-generation DNA sequencing, or possible methodological limitations. However, genetic laboratory methodologies and approaches were cited and discussed by Butler and Duis [92] and Ho et al. [13]. These limitations will be appropriately addressed in future studies, but are beyond the scope of this review.

## 4. Conclusions

Syndromic autism, behavioral disorders, and psychiatric illnesses are highly heterogeneous and are associated with significant health and economic burdens worldwide. Knowledge gaps continue to exist, including a lack of early predictors or biomarkers followed by a paucity of data which contributes to our lack of understanding of the underlying biological mechanisms that lead to comorbidities for many of these conditions. Associated features (which may vary from patient to patient) and identified targeted mechanisms for specific treatments to address the clinical presentations are lacking in most cases of ASD, thus limiting the effectiveness of treatment interventions. Clearly, more research is needed.

The identification of the genotype or genetic finding has the potential to directly influence the patient’s care, quality of life, prognosis, and outcome. To develop a better understanding of the specific gene(s) or chromosomal defects leading to a specific genetic syndrome associated with autism, it is necessary to explore relevant pathways which impact neurodevelopment, behavior, psychosocial functioning, and psychiatric symptomology. When identified, these pathways are often found to be linked to phenotypic variation, neurodevelopmental outcomes, clinical severity, and treatment response. Once novel targets are successfully identified, they provide opportunities leading to clinical trials or treatments with the potential to improve the lives of those with syndromic autism and other neurodevelopmental disorders.

## Figures and Tables

**Table 1 brainsci-14-00343-t001:** Genetic syndromes associated with autism and other neuropsychological, behavioral, and psychiatric disorders.

Genetic Syndrome	Gene(s) or Chromosome Region	Estimated Prevalence	Estimated Rate of ASD	Neuropsychological Correlates	Behavioral Concerns	Psychiatric Disorders
Fragile X Syndrome	FMR1	1 in 4000 male and 1 in 8000 female births	about 50% of males, 20% of females	IQ range from average to severe IDD, with IDD in 85% of males and 25% of females, with a general trajectory of decline in IQ score with increasing age in males, specific learning disorders are common	ADHD, self-injurious behaviors	Generalized anxiety disorder, specific phobia, social anxiety disorder or obsessive–compulsive disorder, depression
Tuberous Sclerosis Complex	TSC1, TSC2	1 in 6000 births	40% to 50%	IQ range from average to severe IDD, with IDD in 56%, with a greater likelihood of IDD with TSC2 mutations, seizures are common	ADHD, aggression, tantrums, eating problems, self-injurious behaviors, insomnia	Anxiety and depressive disorders
Phelan–McDermid Syndrome	22q13.3 del SHANK3	2.5–10 per million births	greater than 90%	Moderate-to-profound IDD is typical, seizures as well as periods of developmental regression or catatonia are common	Stereotypies, aggressive behaviors, recurrent self-injury, and repetitive self-stimulatory behaviors	Anxiety disorders, obsessive–compulsive disorder, depressive disorders, bipolar mood disorder, and schizophrenia spectrum disorders
Prader–Willi Syndrome	15q11-q13 SNRPN	1/20,000–1/30,000 births	about 25%	Mild-to-moderate IDD is typical, those with mUPD compared with DEL have on average higher cognitive skills. Attentional deficits, cognitive rigidity, deficiencies in basic language skills, and impaired visuospatial abilities are common	Hyperphagia-driven behaviors (e.g., food obsessions, hoarding, or foraging), temper tantrums, aggression, self-injurious behaviors, and stubbornness, rigid inflexibility, repetitive, compulsive, ritualistic, or manipulative behaviors	Anxiety, depression, bipolar disorder, schizophrenia, with psychosis including a phenomenon called ‘cycloid psychosis’, much more common in the mUPD genetic subtype
Angelman Syndrome	15q11-q13 UBE3A	1 in 15,000 births	about 30%	Severe IDD and seizures, with maternal deletion of 15q11-q13 most often associated with greater limitations in global development	ADHD, uncontrollable bouts of laughter can escalate into a nearly convulsive state, repetitive and stereotyped behaviors with the potential for causing self-injury	Anxiety including separation anxiety and specific phobias such as fear of noise or crowds
DiGeorge Syndrome	22q11.2 del (dozens of related genes, e.g., TBX1, PRODH, COMT, TUPLE1)	1 in 6000 births	up to 50%	IQ ranges from average to mild IDD, specific learning disorders are common, with familial deletions showing a more severely affected phenotype compared to de novo deletions, seizures and Parkinson disease are common	ADHD, overactive, impulsive, irritable, emotionally labile, rigid, inflexible, shy, withdrawn, aggressive, disinhibited	Anxiety disorders including generalized anxiety disorder, separation anxiety disorder, social anxiety, panic disorder, specific phobias, and obsessive–compulsive disorder, and a remarkably elevated risk for psychosis, with psychotic disorders affecting 10% of adolescents and 30% of adults
Phenyl- ketonuria	PAH	1 in 12,000 births	3% to 6%	IQ scores mostly in average range with early treatment though with increased rates of learning problems. Untreated PKU is characterized by mild-to-profound IDD microcephaly, seizures, motor deficits, developmental delays	ADHD, school problems, decreased motivation, less social competence, irritability, and low self-esteem	Anxiety disorders including generalized anxiety, panic disorder, specific phobias, and obsessive–compulsive disorder, mood disorders including depression, and psychotic disorders including schizophrenia spectrum disorder
Down Syndrome	Trisomy 21	1 in 1000 pregnancies	up to 40%	Most often moderate IDD, with range from mild to severe, with the risk for seizures and early onset dementia increasing with age, with about 15% at age 40 years or older developing Alzheimer disease	Externalizing behaviors (aggression, conduct problems), internalizing behaviors (social withdrawal, secretiveness), impulsivity, attention problems (despite a lower incidence of ADHD per se), irritability, self-control, emotional withdrawal	Anxiety disorders including obsessive–compulsive disorder, mood disorders including depression, and psychotic disorders including schizophrenia, with females having a greater risk for psychotic disorders compared to other cohorts in which there is usually a lack of gender differences reported in psychosis
Rett Syndrome	MECP2	1 in 10,000 female births	about 50%	Low average IQ to severe IDD, early neurological regression that affects motor, cognitive, and communication skills, epilepsy in most, increased risk of sudden death due to autonomic dysregulation	Internalizing (social withdrawal, anxiety, depression, mood lability) and externalizing (aggression), self-injurious behaviors, unprovoked outbursts of screaming, laughing, or uncontrollable crying spells	Social anxiety and severe persistent anxiety with generalized tension and panic attacks, generally more severe in individuals with mild MECP2 pathogenic variants, when compared to participants with either moderate or severe pathogenic variants
Williams Syndrome	7q11.23 del (dozens of related genes, e.g., ELN)	1 in 20,000 births	about 20%	Mild-to-moderate IDD or learning disability, with strength in language, yet extreme weakness in visuospatial construction, problems with sensory integration, adaptive functioning declining over time	Attention problems, poor emotional self-control, low frustration tolerance, temper tantrums, and physical aggression	Social anxiety, separation anxiety, generalized anxiety disorder, obsessive–compulsive disorder, and specific phobias (often relating to noise stimuli and blood, injury, and injections, in contrast to more common phobias reported in typically developing peers)
Burnside–Butler Syndrome	15q11.2 BP1-BP2 del (NIPA1, NIPA2, CYFIP1, TUBGCP5)	Unknown	up to 50%	IQ ranges from average in about one-third to global developmental delay with IDD in two-thirds, increased incidence of memory problems, learning disorders, and seizures	ADHD, oppositional defiant disorder, self-injurious behaviors	Obsessive–compulsive disorder, psychotic disorders including schizophrenia
Cornelia de Lange Syndrome	5p13.2 (5 genes-NIPBL, RAD21, SMC3, HDAC8, SNC1A)	1 in 10,000 to 30,000 births	up to 80%	Range from borderline intellectual functioning to profound IDD, with severe verbal communication deficits. Adaptive behavior is significantly impaired, and tends to worsen with age, more markedly in those with variants in NIPBL. Seizures are common	ADHD, hyperactivity, self-injury, daily aggression, and sleep disturbances, which correlate closely with the degree of learning impairment. Repetitive behaviors including ‘circling, twirling, whirling’ and hand posturing, self-injurious behaviors	Anxiety disorders including social anxiety with selective mutism, obsessive–compulsive behaviors which tend to worsen with age

IQ = intelligence quotient, IDD = intellectual development disability, ADHD = attention deficit hyperactivity disorder, mUPD = maternal uniparental disomy.

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
