# Peer review of "Behavioral and Psychiatric Disorders in Syndromic Autism"

_brainsci, 2024, doi:10.3390/brainsci14040343_

Round 1
Reviewer 1 Report
Comments and Suggestions for Authors
The manuscript presents a broad review of syndromic ASD regarding aspects of genetics, neurodevelopment, and psychiatric presentations. The review has a well-organized and predictable structure, making it easy for the reader to follow each sub-chapter. Please find below my specific comments and suggestions:
- Lines 13-15: I don’t think the review is targeted at caregivers in its present form as little to no emphasis on caregivers is made. I suggest removing this statement from the abstract.
- Sub-chapter 2.6: You interchangeably use DGS and VCFS without a clear reason. Additionally, the full form of the VCFS abbreviation is lost at line 480. I suggest using only one of the terms to not confuse the reader.
- Table 1: In the DiGeorge Syndrome row there is a “familial” with an italic “a” character.
- Pharmacogenomics is briefly discussed in the last paragraph of the Discussion. However, except for the Prader-Wili syndrome, I don’t think the topic was properly reviewed in the manuscript. Please add more elements of pharmacogenomics to the review and expand the discussion on this or I suggest removing it.
Author Response
Authors responses noted in RED
Reviewer 1
The manuscript presents a broad review of syndromic ASD regarding aspects of genetics, neurodevelopment, and psychiatric presentations. The review has a well-organized and predictable structure, making it easy for the reader to follow each sub-chapter. Please find below my specific comments and suggestions:
- Lines 13-15: I don’t think the review is targeted at caregivers in its present form as little to no emphasis on caregivers is made. I suggest removing this statement from the abstract.
Agree. Removed “caregivers” from abstract.
- Sub-chapter 2.6: You interchangeably use DGS and VCFS without a clear reason. Additionally, the full form of the VCFS abbreviation is lost at line 480. I suggest using only one of the terms to not confuse the reader.
Agree. Now using DGS abbreviation exclusively.
- Table 1: In the DiGeorge Syndrome row there is a “familial” with an italic “a” character.
Correction made.
- Pharmacogenomics is briefly discussed in the last paragraph of the Discussion. However, except for the Prader-Wili syndrome, I don’t think the topic was properly reviewed in the manuscript. Please add more elements of pharmacogenomics to the review and expand the discussion on this or I suggest removing it.
We would opt to keep this emerging topic in the review and the authors consider the mention and added brief description of pharmacogenomics in the Discussion section as important and is becoming an essential element to evaluate and develop treatment plans likely involving psychopharmacology in behavioral and psychiatric disorders frequently associated with these autism syndromes. We did, however, modify the organization of the Discussion section accordingly.
Reviewer 2 Report
Comments and Suggestions for Authors
In this article entitled "Behavioral and Psychiatric Disorders in Syndromic Autism" the authors examine syndromic autism in detail, highlighting the clinical features, behavioural traits and genetic correlates associated with this complex condition. The article provides a comprehensive overview of syndromic autism, including providing demographic data, exploring technological advances in genetic diagnosis, and describing clinical and therapeutic implications.
The notable concerns I have are as follows:
First, although the article provides a detailed introduction to syndromic autism, it seems to focus more on describing general characteristics and demographics than on providing an in-depth analysis of the underlying biological mechanisms. This approach limits readers' understanding of the biological processes involved in this condition.
Secondly, the article lacks discussion of theoretical approaches that might explain the links between specific genetic mutations and the behavioural manifestations of syndromic autism. Further exploration of these perspectives would have provided a sound conceptual framework for interpreting the results of the study.
Thirdly, although the article refers to technological advances such as next-generation sequencing, it does not discuss the possible methodological limitations of these approaches. A discussion of these aspects would have provided a fuller context for interpreting the results of the study and would have strengthened its credibility.
Finally, the article briefly mentions the importance of accurate identification of genes or chromosomal abnormalities to guide treatment, but does not provide an in-depth discussion of specific therapeutic approaches or interventions available for patients with syndromic autism.
What are the limitations of this study?
Author Response
Authors responses noted in RED.
In this article entitled "Behavioral and Psychiatric Disorders in Syndromic Autism" the authors examine syndromic autism in detail, highlighting the clinical features, behavioural traits and genetic correlates associated with this complex condition. The article provides a comprehensive overview of syndromic autism, including providing demographic data, exploring technological advances in genetic diagnosis, and describing clinical and therapeutic implications.
The notable concerns I have are as follows:
First, although the article provides a detailed introduction to syndromic autism, it seems to focus more on describing general characteristics and demographics than on providing an in-depth analysis of the underlying biological mechanisms. This approach limits readers' understanding of the biological processes involved in this condition.
Our response to this critique is that a description of the biological processes is beyond the scope of this article and may stimulate additional studies in the future and a review of biological processes and/or gene-gene or protein interactions in syndromic autism.
Secondly, the article lacks discussion of theoretical approaches that might explain the links between specific genetic mutations and the behavioural manifestations of syndromic autism. Further exploration of these perspectives would have provided a sound conceptual framework for interpreting the results of the study.
Our response to this critique is that an explanation of the underlying mechanisms involved which link the genetic mutations to the behavioural manifestations of syndromic autism is beyond the scope of this article with the same reasoning as listed above and would require a separate review article in syndromic autism.
Thirdly, although the article refers to technological advances such as next-generation sequencing, it does not discuss the possible methodological limitations of these approaches.
Genetic laboratory methodology and approaches were cited in this review (Butler and Duis-reference 92 and by Ho et al -reference 13) for the readership to pursue more information on this topic and beyond the scope of this review and would be a topic for a separate report as the field of genomic technology and methods have changed significantly in the past few years impacting the identification genetic causes of ASD and syndromic autism.
Finally, the article briefly mentions the importance of accurate identification of genes or chromosomal abnormalities to guide treatment but does not provide an in-depth discussion of specific therapeutic approaches or interventions available for patients with syndromic autism.
Although specific treatments are beyond the scope of this article, we do mention treatment implications in the discussion section, as follows: “Pharmacogenomics is an emerging field in medical care and clinical practice that can impact treatment and medical care. Evidence based medication selection and dosing strategies may be driven by the pharmacogenetics or DNA pattern of liver enzymes involved in drug metabolism which are unique to each individual patient. The use of this precision medicine approach in treating those affected with neurodevelopmental disorders including ASD will be important, as evidenced by the report from Forster et al. in 2021 while treating aberrant behavior in a patient with Prader-Willi syndrome [232]. Depending on the clinical presentation, other diagnostic genetic laboratory testing may be considered including gene repeat expansion analysis, e.g., FMR1 gene for fragile X syndrome [25], high-resolution chromosome microarray studies [13], or other laboratory methods such as mitochondrial testing [12]. Laboratory based methods are generally excellent in identifying genetic defects associated with specific identifiable biological markers that can be evaluated and followed, with consideration of potential treatment avenues which might be available or discovered over time.”
What are the limitations of this study?
Agree. We have added mention of the first and second critiques in a paragraph added to our discussion summarizing the limitations of this study. Mention of the final critique is added as the last sentence in the Introduction section.
Reviewer 3 Report
Comments and Suggestions for Authors
People with Williams syndrome do not have autistic features, neither do people with ANgelman syndrome.
Men with fragile X syndrome have autistic features, but do not fill the criteria of autism spectrum syndrome.
Author Response
Authors responses noted in RED.
People with Williams syndrome do not have autistic features, neither do people with ANgelman syndrome. Men with fragile X syndrome have autistic features, but do not fill the criteria of autism spectrum syndrome.
We acknowledge that the “typical” cases of Williams, Angelman or Fragile X syndrome do not meet criteria for autism. However, references (e.g.,4,185,186, 188) provide evidence for the 20% of individuals with Williams syndrome who meet criteria for ASD, which is well above population norms. Reference 94 does the same for Angelman syndrome, for which the incidence of ASD is 34% and the UBE3A gene involved in Angelman syndrome is a recognized autism gene (reference 4), and in men with Fragile X syndrome the incidence of ASD is about 50%, supported by reference 27.